# The Many Faces of Cyclodextrins within Self-Assembling Polymer Nanovehicles: From Inclusion Complexes to Valuable Structural and Functional Elements

**DOI:** 10.3390/ijms25179516

**Published:** 2024-09-01

**Authors:** Ivana Jarak, Sara Ramos, Beatriz Caldeira, Cátia Domingues, Francisco Veiga, Ana Figueiras

**Affiliations:** 1Laboratory of Drug Development and Technologies, Faculty of Pharmacy, University of Coimbra, 3000-548 Coimbra, Portugal; jarak.ivana@gmail.com (I.J.); sararamos2002@gmail.com (S.R.); bea.caldeira2@gmail.com (B.C.); cdomingues@ff.uc.pt (C.D.); fveiga@ff.uc.pt (F.V.); 2Instituto de Investigação e Inovação em Saúde, University of Porto, 4200-135 Porto, Portugal; 3REQUIMTE/LAQV, Group of Pharmaceutical Technology, University of Coimbra, 3000-548 Coimbra, Portugal; 4Institute for Clinical and Biomedical Research (iCBR), Area of Environment Genetics and Oncobiology (CIMAGO), Faculty of Medicine, University of Coimbra, 3000-548 Coimbra, Portugal

**Keywords:** amphiphilic cyclodextrins, self-assembly, drug delivery systems, nucleic acids, proteins

## Abstract

Most chemotherapeutic agents are poorly soluble in water, have low selectivity, and cannot reach the tumor in the desired therapeutic concentration. On the other hand, sensitive hydrophilic therapeutics like nucleic acids and proteins suffer from poor bioavailability and cell internalization. To solve this problem, new types of controlled release systems based on nano-sized self-assemblies of cyclodextrins able to control the speed, timing, and location of therapeutic release are being developed. Cyclodextrins are macrocyclic oligosaccharides characterized by a high synthetic plasticity and potential for derivatization. Introduction of new hydrophobic and/or hydrophilic domains and/or formation of nano-assemblies with therapeutic load extends the use of CDs beyond the tried-and-tested CD-drug host–guest inclusion complexes. The recent advances in nano drug delivery have indicated the benefits of the hybrid amphiphilic CD nanosystems over individual CD and polymer components. This review provides a comprehensive overview of the most recent advances in the design of CDs self-assemblies and their use for delivery of a wide range of therapeutic molecules. It aims to offer a valuable insight into the many roles of CDs within this class of drug nanocarriers as well as current challenges and future perspectives.

## 1. Introduction

Since the discovery of cyclodextrins (CD) in 1891 by the French Antoine Villiers, numerous studies have been carried out to investigate their synthetic nature, structure, and properties. Cyclodextrins are well-known excipients used in several clinically approved formulations [1]. They are biocompatible and biodegradable cyclic oligosaccharides with hydrophilic exteriors and hydrophobic cavities which can be used to encapsulate hydrophobic molecules. The inclusion of chemotherapeutics is widely studied as the means of improving dissolution rate, physicochemical stability, bioavailability, and therapeutic duration. The presence of the outer hydroxy groups enables derivatization which expands the usefulness of cyclodextrins as carriers of therapeutic molecules. CDs can be used as components of delivery systems as functional drug reservoirs/linkers/hydrophilic tails, and examples where CDs are used to functionalize self-assembling polymers, solid/metal nanoparticles, or dendrimers are regularly compiled and updated [2,3,4,5,6]. Alternatively, CDs can be functionalized into amphiphilic molecules with the property of self-assembly [7] or can self-assemble into nanocomplexes via non-covalent interactions with therapeutic natural polymers like proteins and nucleic acids. The synthetic plasticity of CD faces can be complemented with the insertion of other functionalized ligands and direct assemblies’ morphology, physicochemical properties, and applicability as vehicles of therapeutic molecules. For example, the addition of cationic components to amphiphilic CDs provides the opportunity to bind different types of therapeutic nucleic acids and proteins through electrostatic interactions. Compartmentalization of hydrophilic, hydrophobic, and host components can be exploited for delivery of hydrophilic nucleic acids and proteins, hydrophobic compounds, or their combinations. However, the attempts to optimize the structure–activity relationship so far have been few and contained only a small library of derivatized CDs, leaving room for further optimization of physicochemical and therapeutic parameters. Additionally, most of these attempts were oriented towards the delivery of hydrophobic drugs and their combinations. Although self-assemblies of amphiphilic CD are increasingly being explored as carriers of hydrophilic therapeutics (nucleic acids and proteins), the opportunity the hydrophobic pocket provides for the inclusion of hydrophobic chemotherapeutics/imaging molecules tends to be neglected, and examples of drug/NA co-delivery by functionalized self-assembling CDs are still rare. Finally, the ability of CDs to thread amphiphilic polymers results in self-assembling polyrotaxane species with unique pharmacological properties.

The tremendous potential of CD to interact with hydrophobic moieties via host–guest inclusion into the hydrophobic CD pocket, the selective polymer threading potential, and the huge number of sites for potential chemical derivatization has made the applicability of CD in the universe of polymer-based drug nanovehicles almost overwhelming. Among them, amphiphilic CDs present a class with increasing potential for delivery of diverse kinds of therapeutic molecules. Interestingly, the same formulation often presents universal capacity for delivery of various classes of therapeutics. In this review paper, we offer an overview of the progress achieved in the niche of amphiphilic CDs in this decade, with a special overview of potential development and future directions.

## 2. General Considerations on Amphiphilic Cyclodextrins

CDs are macrocyclic oligosaccharides consisting of α (1→4)-linked D-(+)-glucopyranosyl units in the form of a truncated cone with a hydrophilic outer surface and a hydrophobic cavity (Figure 1).

This hydrophobic cavity can be attributed to the fact it is coated with skeletal carbons and ether oxygens from the pyranose configuration of the glucose molecules [8]. The polarity of the cavity has been estimated to be similar to that of an aqueous ethanolic solution [9]. In the circular arrangement of glucopyranosides, the hydroxyl groups are exposed on opposite sides of the truncated cone. The primary 6-hydroxy groups are located at the narrower edge and the secondary 2- and 3-hydroxy groups at the wider edge. There are three natural or parental CDs of pharmacological relevance characterized by different numbers of D-(+)-glucopyranosyl units and consequently in their size: αCD has six units, βCD seven, and γCD eight. The βCD is the most-used and -produced CD in the pharmaceutical industry. This fact can be attributed to its ease of preparation, efficient drug complexation, availability, low cost, and the ideal cavity size [10]. βCD’s geometry is also the most appropriate for encapsulation, since αCD’s cavity is not large enough to accommodate most of drugs and γCD has a large cavity resulting in unstable interactions during complexation. NMR and diffraction experiments demonstrated that the high crystal lattice energy and intramolecular hydrogen bonds prevent CDs from interacting with the neighboring water molecules and render all natural CDs only partially soluble in water [11]. To overcome this problem, chemically modified derivatives with better inclusion capacity and water solubility have been synthesized. 

Polymers are widely used for CD modification and in the formation of various types of nanocomplexes with cyclodextrins. It is therefore important to know the different factors that play a role in the formation of such nanocomplexes.

Derivatization of multiple CD sites on either or both faces gives rise to star-shaped polymers in which CD serves as a branching node. Compared to linear counterparts of the same molecular weight, star-shaped polymers provide several advantages as nanovehicles. A high degree of substitution results in assemblies of lower hydrodynamic radius with lower solution viscosity and higher density of exposed functional groups. Additionally, drug encapsulation and loading capacity are higher than for linear polymers. In some cases, they form unimolecular micelles resistant to disassembly upon dilution [12,13,14].

The shape and size of self-organized grafted βCDs can depend on the chemical nature of grafted polymers, the position of substituents, or a degree of substitution. Nonetheless, the presence of three types of hydroxylic groups on glucopyranose rings (secondary OH in positions C2 and C3 and primary OH on C6) with distinct reactivities poses special demands on the synthetic process and dictates the isomer composition, substitution site, and substitution degree of CD–polymer conjugates. Over the past two decades, the growing interest in amphiphilic CDs has resulted in increased optimization in CD derivatization. A variety of persubstituted (all primary and/or secondary OH groups substituted) and randomly and selectively substituted CDs were prepared and used as a starting point for the synthesis of amphiphilic CDs [15,16].

Acylation by fatty acids is one of the earliest modifications used to prepare amphiphilic CDs. The stability of self-aggregated persubstituted CDs acylated on the secondary side is heavily dependent on the presence and the nature of substituents introduced into the primary face of the CD ring [17,18]. In the case when there are no substituents on the secondary face of acylated CDs, stable aggregates are produced only by a mixture of incompletely substituted molecules. A more recent study of self-assembling behavior of a βCD library obtained by a straightforward thermolysin-biocatalyzed acylation revealed the influence of the alkyl chain length and total degree of substitution (TDS) on the morphology of supramolecular structures produced by nanoprecipitation (Figure 2A) [19,20]. Deep insight into these nanostructures provided by the high-resolution cryo TEM demonstrated the formation of spherical multilamellar nanoparticles by the βCDs with TDS < 5 (Figure 2B). For TDS > 5, a variety of shapes was observed, including barrels and highly irregular shapes. In both cases, the morphology seems to be mainly guided by the degree of substitution and not the chain length. Most of the glucose units were acylated in the C2 position by this synthetic method, but with a poor reproducibility of substitution degree. Similar nanostructure morphologies were observed when acylated isomers with controlled acyl chain distribution were prepared by a more reproducible method of trans-esterification with vinyl esters in a presence of a mild mineral base catalysis [21].

Two major routes towards CD–polymer amphiphiles include the grafting of a CD core with already-made polymers (“arm-first”). “Core-first” is associated with the use of living polymerization methods and relies on using CD as an active core upon which polymer chains are gradually extended. Some of the most-applied methods for preparing “core-first” CDs include atom transfer radical polymerization (ATRP) [22,23], reversible addition-fragmentation chain-transfer (RAFT) polymerization [24], and ring-opening polymerization [25].

Apart from derivatization of the CD core by sequential precursor deprotection and chemical grafting [26], mixed functionalized CD nanosytems can be obtained during the self-aggregation step. Hydrophilic polyethylene glycol (PEG) and its alternatives have become established structural elements for improving the stability and overall performance of drug nanocarriers [27]. They form hydrophilic shields of adjustable topologies which inhibit interactions with plasma proteins and subsequent clearance by the mononuclear phagocyte system. Therefore, they can prolong systemic circulation of nanovehicles, potentiate increased accumulation at the diseased site via passive mechanisms, and mitigate loss of therapeutic load or nanovehicle toxicity.

However, PEGylation can be associated with immunogenicity [28], loss of cargo [29], or loss of therapeutic efficacy [30]. For example, a mixed nanosystem composed of PEGylated and cationic amphiphilic CDs designed for siRNA delivery displayed improved stability at a cost of reduced cell uptake and gene silencing [30]. Although the advantages of PEGylation are well documented, the influence of PEG topology on the properties of nanosystems described in this review paper is less investigated. Nonetheless, it is reasonable to extend the general considerations of PEGylation to CD-based nanosystems as well [31,32]. Therefore, the grafting density and molecular weight of PEG chains have to be carefully considered and should be a part of the rational design of a nanocarrier [33]. Similarly, simultaneous addition of emulsifiers or solubilizers is a tested approach to improve the pharmacokinetic properties of nanoparticles [17].

Postinsertion, which is often used for decoration of liposomes or cell membrane vesicles, can also be used in the case of CD vesicles. Insertion of 1,2-diacyl-rac-glycero-3-methoxypolyethylene glycol-2000 (DMG-PEG2000) is a common way to PEGylate amphiphilic CDs [21]. Finally, the hydrophobic cavity of a CD allows the insertion of functionalized hydrophobic linkers and provides an additional site for functionalization [34].

The same approaches can also be used for introducing targeting ligands. Folic acid, galactosyl or lactosyl groups, the Arg-Gly-Asp (RGD) sequence peptides, transferrin, hyaluronic acid (HA), and monoclonal antibodies are widely used as targeting ligands to increase the nanoparticles’ ability to target the cancer cells after accumulation in tumoral mass by a passive enhanced permeability and retention mechanism (EPR) [35,36,37,38].

The final tool in the arsenal of smart nanovehicles is ensuring the controlled release of therapeutic cargo in a specific tissue or cellular compartment. Particular characteristics of targeted sites are used to design responsive structural motifs that are incorporated within the polymer structure. Under stimuli-provided intracellular, extracellular, or external queues, they induce nanoassembly disintegration and release of therapeutic load. Some of the exploited structural features respond to internal pH, redox, or enzymatic stimuli, while others are triggered by external light, ultrasound, or temperature stimuli.

CD-threaded polymers hold great potential in various fields, ranging from new material design to application in medical and pharmaceutical field. Nonetheless, the applicability of polyrotaxanes (PR) based on non-substituted CDs is limited by water insolubility brought about by strong hydrogen bonds between threaded CDs. Nonetheless, water solubility and self-assembly can be controlled by the degree of CD substitution and the size of the polymer backbone, as was recently described for acylated αCD and high-molecular weight PEG PRs [39]. Under appropriate conditions, hydrophobic domains of substituted CDs drive the formation of nanoparticles. However, optimization of polymer properties and the nature of CD substituents is indispensable for the formation of stable water-soluble nanoaggregates. The choice of end-cap bulky stoppers helps fixate the inclusion complex structure and can be designed to promote drug release in a stimuli-controlled manner, as will be described further on [40].

On the other hand, derivatization of CD hydroxy groups can enable the complexation of charged hydrophilic molecules like nucleic acids. The threading efficiency of the polyester backbone was the driving force for efficient transfection with silencing RNA (siRNA) complexed by *N*,*N*-dimethylethylenediamine-derivatized αCDs [41]. The nature of the backbone also proved decisive for the threading, and the polyesters were more efficient than PEG. Additionally, the sizes of the monomers (number of ester groups) and the axel also influenced coiling and aggregation properties of the produced PRs, and a hollow spherical morphology was observed due to hydrogen bond interactions between the unsubstituted hydroxy groups of the CDs [41]. A series of studies undertaken by Taharabaru et al. determined the importance of fine-tuning of amino group modifications, degradable linkers between the endcaps and polymer axel, the degree of amine substitution and CD threading, and the molecular weight of the axel in creating adjustable molds for highly complex therapeutic species like Cas9/single guide RNA [42,43,44].

The structural properties of polymers can be combined with the size of CD cavities to guide the position of threading. In Poloxamers composed of the central polypropylene (PPO) block flanked by PEG chains, αCDs favor threading of PEG units while βCDs are colocalized on the PPO center [45].

In PRs, stimuli-sensitive groups are usually introduced at the ends of polymer axels. Upon application of stimuli, the endcaps are dissociated from the axel and enable the release of threaded CDs. This approach was used by Tamura et al. for delivery of CD-conjugated drug to cancer cells [46]. More controlled light-triggered conditions were used for the release of the drug entrapped within the hydrophobic domains of propionylated PR end-capped with UV-sensitive nitobenzyl stoppers [40,47]. Similarly, targeting moieties can also be introduced by CD derivatization [48]. PRs are usually prepared by CD threading on already available polymers. However, functionalized poly(pseudo)rotaxanes (PPR) can also be grafted on complementary polymer chains. For example, PEG-based PPR was grafted on poly(ε-caprolactone) of appropriate size while the other end was stopped with folic acid [49]. In the mixed nanoparticles prepared from mPEG_1k_-b-PCL_4k_ and FA-PEG_1.5k_-b-PCL_4k_ PR, the NMR experiments demonstrated that threaded CD influenced PEG conformation and improved the exposure of FA on the surface of nanoparticles when compared to formulations without CDs and was reflected in cell uptake [49].

As demonstrated in some of the above examples, there is evidence that PRs selfassemble into a variety of morphologies. Nonetheless, in this class of CD derivatives, the authors seem to be more focused on therapeutic performance, at least based on the examples published in this decade. To keep within the scope of this review which is dedicated to the polymer-based selfassembling CDs, in the following chapters we have included only those examples which have undertaken studies of polyrotaxane nanoassembly size or morphology.

Unlike the end-capped PRs, PPR are more considered in the scope of drug delivery vehicles based of the higher order of assemblies like hydrogels or nanosheets which are beyond the scope of this review [50,51]. Nonetheless, PPRs formed by di-block co-polymers like methoxy poly(ethylene glycol)-poly(l-lactide-*co*-glycolide) (mPEG-PLGA) threaded by αCD form nanoparticles that were successfully applied to drug delivery [52].

We shall demonstrate how the principles of nanosystem design are applied to selfassembling amphiphilic and threaded CDs in the following chapters.

## 3. Drug Delivery

The most recent examples of drug delivery by the CD-based self-assemblies are presented in Table 1. 

To improve stability and uncontrolled drug release, Hong et al. constructed folate (FA)-conjugated βCD-polycaprolactone (βCD-PCL) copolymers loaded with curcumin. In this study, βCD was modified with biocompatible and biodegradable PCL to adjust the hydrophilic–lipophilic balance, consequently controlling the release rate of the drug [53]. βCD-PCL copolymer was synthesized by ring-opening polymerization of caprolactone monomers in the presence of βCD using tin octoate as a catalyst. The copolymer was then conjugated with FA to achieve a targeting effect on cancer cells and was formulated as nanoparticles by emulsion-solvent evaporation. The obtained nanomicelles were stable under the normal physiological conditions and displayed steady drug release over five days after the initial burst. Additionally, the release rate at the tumor site was three times higher than in the circulatory system owing to the acid-sensitive ester bridges. Furthermore, targeting ligands significantly improved in vitro and in vivo therapeutic outcomes. 

The two-stage drug release is often observed in micelles containing CDs and hydrophobic block copolymer. Molecular docking demonstrated that in the case of βCD and hydrophobic cholesterol (Chol) linked by hydrophilic PEG, the curcumin release profile could be explained by inclusion into the hydrophobic cavity of CD, hydrophobic interactions with hydrophobic Chol core, and the hydrogen bonds [54]. Although prolonged sustained in vitro release has been observed in the above examples under physiological conditions, there is a concern that it might be related to the development of multiple drug resistance.

Dox is widely used in clinical practice. However, it has poor water solubility, short residence time in the circulation, inefficient biodistribution, and many associated adverse effects. To overcome these problems, it is often assembled into polymeric nanoparticles. Previous studies on star CDs substituted with branched amphiphilic dextran (Dex) revealed the influence of the CD degree of substitution (DS) on micellar stability [55]. However, anticancer activity of micellar Dox was lower than that of free Dox despite cell internalization and cytoplasmic colocalization of micelles, presumably due to intense binding of Dox within the hydrophobic core of star micelles. Alternative star copolymer micelles for encapsulation of superparamagnetic iron oxide (SPIO) and Dox were prepared by grafting PCL to the secondary face and complementing it with the hydrophilic dextran on the primary face [56]. To improve cancer cell targeting, dextran was decorated with FA. In both examples, the bulkiness of Dex dictated the DS on the primary face. While nanomicelles linked by disulfide bridges rapidly released Dox in the presence of a high concentration of GSH and enabled its colocalization within the nucleus, free Dox exhibited higher therapeutic effects within the limited scope of in vitro experiments. Although promising, the true potential and advantages of these theranostic micelles should be tested in vivo.

Combined chemotherapy has been a viable option for the treatment of cancer by exploiting additive or synergistic effects of complementary therapeutic. Rahmani et al. developed a new pH-sensitive micelle composed of biodegradable βCD grafted with amphiphilic poly maleate-block-PLGA (βCD-g-PMA-co-PLGA) for codelivery of doxorubicin (Dox) and adjuvant conferona (Con) [57]. The combination of different binding mechanisms (electrostatic/hydrophobic for Dox and hydrophobic/drug–host for Con) and hydrolysis of pH sensitive hydrophobic PLGA micellar core resulted in sustained release of both drugs (Figure 3). Although formed micelles had very low cmc (<3 µg/mL) similar to other star-shaped copolymer CD micelles, high PDI possibly reflected reduced stability or heterogeneity in polymer composition. Nonetheless, they were efficiently internalized by cancer cells and exerted synergistic antiproliferative activity against the metastatic breast cancer cells.

A dense, highly branched structure of dendrimers emerged as an ideal carrier for nitric oxide (NO), a potent antibacterial agent. However, despite their high capacity for NO, the application of higher generations of poly(amidoamine) dendrons (PAMAM) is limited by cytotoxicity. The formation of star copolymers emerged as a solution to mitigate the observed toxicity and improve functionality of dendrons. Liu et al. designed a biocompatible star copolymer βCD-PAMAM by uniform substitution of primary CD hydroxyls via azide-based click chemistry [58]. NO was loaded by the conversion of secondary PAMAM nitrogens into N-diazeniumdiolate (NANOate), with the drug loading capacity superior to that of small molecular vehicles. Gradual release of NO from CD-dendrimer led to degradation of existing and inhibited formation of new bacterial biofilms, both in vitro and in vivo after intranasal spray administration.

Polycationic CD-based brushes have been developed as a new type of carrier material of antimicrobial agents for phototherapy of periodontitis. A star-shaped polycationic brush of poly(2-(dimethylamino)ethyl methacrylate) (PDMA) was synthesized via atom transfer (ATRP) of DMA monomer from the initiator bromo-substituted CD [59]. The high grafting density of PDMA (βCD-PDMA_7_) forced the extended cationic chains into a brush conformation as evaluated from the distance between the neighboring chains and the gyration radius (D/Rg < 2). The negatively charged ICG was then loaded into βCD-PDMA_7_ via combination of polyionic complex with PDMA and host–guest interactions with CD. The positively charged hydrophilic face of βCD-PDMA_7_@ICG significantly promoted the adsorption and penetration of ICG into bacterial cells and showed excellent results in phototherapy with antibacterial and antiperiodontitis action, effectively inhibiting the resorption of alveolar bone and relieving inflammatory reactions after laser irradiation.

Strong immunosuppressive nature of colorectal (CRC) tumor microenvironment (TME) poses grave impediments towards the wider use of clinically available immunotherapies. Immunogenic cell death (ICD) emerged as a potential mechanism to reverse immunosuppressive TME and can be potentiated by production of ROS. Cationic amphiphilic heptakis [2-O-(N-(3″-aminopropyl)-1′H-triazole-4′-yl-methyl)-6-dodecylthio]-β-CD (NH_2_Pr-CD-C_12_) has previously been used as nucleic acid carrier [60,61]. The size of the hydrophobic chain directs the formation of bilayer vesicles which can be loaded with hydrophobic molecules while positively charged hydrophilic exterior facilitates cell uptake. To improve the stability and pharmacokinetic profile of vesicles under physiological conditions, PEGylation can be introduced by employing various strategies. They include covalent PEGylation, host–guest inclusion or self-aggregation with similar PEGylated species [60,61]. Alternatively, PEG was introduced into a NH_2_Pr-CD-C_12_ vesicle by postinsertion of PEGylated 1,2-distearoyl-sn-glycero-3-phosphoethanolamine (DSPE-mPEG_2k_) [62]. To exploit the synergistic influence of ICD-inducing ginsenoside Rg3 and ROS-inducing quercetin on CRC proliferation, targeting ligand FA was also introduced [62]. FA-decorated formulation improved the pharmacokinetic profile of both drugs and increased tumor accumulation. When administered in vivo, a co-loaded targeting formulation induced infiltration of immunostimulatory cells and enhanced therapeutic outcomes of anti-PD-L1 immune therapy. The same formulation was used for targeted delivery of melasoprol to hepatocellular carcinoma cells [63].

Nanostructured cationic polymers demonstrated great potential in treatment of bacterial infections via inherent antimicrobial activity and ability to incorporate antibiotics. However, lack of cell selectivity often leads to undesirable side-effects. Nanorods can exert greater damage to cells than spheric aggregates due to extended length in one dimension. Glyconanorods constructed by amphiphilic βCD-based glycoconjugates (R_7_-CD-Man_14_) bind to *E. coli* by multiple mannose–protein interactions [64]. Nanorod formation was driven by the combination of hydrophobic and π–π interactions. ATP-rich bacterial environment triggered rhodamine spirolactam opening and fluorescent bacteria sensing accompanied by release of encapsulated antibiotic norfloxacin. Additionally, positively charged ethyleneimine chains of nanorods complemented antibacterial activity. Interestingly, positively charged R_7_-CD-Man_14_ demonstrated insignificant hemolytic effect and toxicity towards healthy cells when compared to the counterpart without mannose substituents (PEG based).

CD-based guest-host complexes can also self-aggregate into drug-loaded nanosystems. Numerous studies have shown that hypoxia is a prevalent phenomenon in solid tumors associated with overexpressed transmembrane protein carbonic anhydrase (CAIX) due to rapid proliferation of tumor cells and tumor heterogeneity. Therefore, CAIX inhibitors, like sulfonamide structures containing benzene or heterocyclic rings, can also inhibit tumor growth and metastasis. It motivated Zhou et al. to develop bilayered supramolecular nanoparticles (SNP) by combining a sulfonamide-coupled mono-(6-deoxy-6-ethyl-enediamino)-CD host with a methotrexate prodrug (Mtx-SS-Ad) [65]. The introduction of amino chains into βCDs resulted in a dramatic improvement in the water solubility of native βCDs and enabled further functionalization. This approach was used to label amino-βCD with L-ligand (sulfonamide) for efficient CAIX inhibition and cancer cell internalization. GSH-sensitive methotrexate-amantadine (Mtx-SS-Ad) prodrug was used to produce an amphiphilic drug–host complex via inclusion of Ad, which self-assembled into nanoparticles in aqueous medium by nanoprecipitation method. After internalization by the tumor cells, the acidic environment and the presence of glutathione lead to the disintegration of the SNP, promoting drug release. Therefore, this approach effectively inhibited the proliferation of cancer cells while minimizing the adverse effects on normal cells.

Additionally, guest–host complexes represent a more recent strategy to impart stimuli-sensitive function to self-assembling nanostructures. They are relatively easy to prepare and result in controllable surface topology. The stimuli-sensitive nature of such complexes is strongly related to the factors that influence their kinetic and thermodynamic properties. Apart from the nature of the host and guest entities, these factors also include external stimuli. The use of CD inclusion complexes also provides a unique opportunity to introduce star-like side chains into non-covalent graft polymers. Adeli et al. prepared poly(2-hydroxyethylmethacrylate)-graft-[polycaprolactone-benzimidazole:β-cyclodextrin-star-poly(methacrylic acid)-block-poly(N-isopropylacrylamide)] [PHEMA-g-(PCL-BM:βCD-star-PMAA-b-PNIPAM)] (*w*/*w* 1:3) micelles by inclusion of hydrophobic benzimidazole-capped PHEMA-g-PCL into hydrophilic βCD-(PMAA-b-PNIPAM)_7_ for improved Dox activity [66]. Modification of thermosensitive PNIPAM block with PMAA and the inclusion of hydrophobic benzimidazole influenced the thermo-responsive behavior of the whole graft polymer, and the hydrophobic character led to PNIPAM collapse and drug release at temperatures characteristic for cancerous tissue (above 40 °C). On the other hand, protonation of PMAA and benzimidazole at lower pH characteristic for endosomal compartment (pH 4.5–6) led to micelle disintegration. In this study, micelles were loaded with Dox HCl by combination of hydrophobic (PHEMA-g-PCL) and electrostatic (PMAA) interactions with the hydrophobic core and hydrogen bonding with the hydrophilic shell (PNIPAM).

Similar concept was exploited for delivery of model drugs by redox and photodegradable 4-methyl-ε-PCL (MPEG_45_-Fc/βCD-ONB-PMCL_23_) [67]. Here, hydrophobic ferrocene (Fc) was used as a guest due to strong interactions with CD. Oxidation of Fc by tumoral H_2_O_2_ leads to controlled degradation of CD-Fc complex and can be exploited for controlled drug release. Complementary spatiotemporal drug release is provided by a photosensitive 2-nitrobenzyl (ONB) linker.

Fc:CD inclusion complex was also used to deliver photosensitizer with a controllable on-off activity, thereby reducing off-target effects. Xue et al. developed a new type of stimuli-sensitive supramolecular complex in which self-aggregation of inclusion complex between phthalocyanine-CD conjugate (Pc-CD) and PEGylated Fc (Fc-PEG) leads to the quenching of the photosensitizer supported by photoinduced electron transfer from Fc [68]. Next, the external stimuli is introduced by targeted delivery of adamantan ligands which have higher affinity for CD and displace Fc from the inclusion complex. Once the amphiphilic character of the micelles is disrupted, photosensitizer is activated and ready to produce reactive oxygen species (ROS). Subcellular localization demonstrated that Pc activation mainly takes place in the endosomal compartment.

The ability of amphiphilic CDs to form bilayered vesicles was exploited to form hybrid dandelion-like vesicles with hydrogenated soybean phosphotidylcholine (HSPC) (Figure 4) [69]. The central core was composed of mixed βCD-NHC_12_H_25_/HSPC vesicle and was loaded with photosensitizer chlorin e6 (Ce6). Epitaxial growth of the dandelion structure was achieved by successive inclusion of a camptothecin (Cpt) prodrug in which Cpt is tethered to CD via a disulfide linker (CPT-SS-CD). Finally, a cancer-cell-targeting peptide was introduced as CPT-SS-cRGDfk. The addition of targeting layer improved dandelion stability and ROS production upon NIR irradiation when compared to free Ce6. Depletion of intracellular GSH for Cpt release additionally boosted the synergistic contribution of produced ROS to the Cpt anticancer activity in vitro and in vivo. 

In most of the host–guest graft polymers CD plays the role of a linker. Nonetheless, it can also serve as an adjuvant. This dual role of CD was described by Chen et al. which prepared supramolecular assembly composed of hexylimidazolium-derivatized CD (AM7CD) and adamantyl-decorated hyaluronic acid (HAAd) [70]. Here, biodegradable HA provides cancer cell targeting as well as drug release upon degradation by tumoral hyaluronidase. Released AM_7_CD has high affinity for intracellular ATP by combining ATP inclusion and electrostatic interaction with the imidazolinium moieties. The AM_7_CD-HAAd successfully enhanced the antiproliferative activity of chlorambucil in drug-sensitive cells. Although not tested in a drug-resistant cell line, the authors hypothesized that the observed AM7CD affinity towards ATP might improve biological activity of anticancer drugs by obstructing the normal function of ATP-dependent drug efflux pumps.

Apart from participating as the host–guest linker or a core for covalently grafted star polymers, CDs can also serve as hydrophilic copolymers to improve the solubility and stability of the functional hydrophobic cores. Unlike classical fluorescent imaging agents that undergo aggregation-caused quenching, some molecules benefit from motion restraint and exhibit aggregation-induced emission (AIE). AIE tetraphenylethylene 1, 2-diphenyl-1,2-(p-hydroxyphenyl)-ethylene (OH-TPE-OH) bridged by polyurethane hexamethylene bridges was end-capped with βCD to produce selfaggregating theranostic nanoparticles (CD-TPE) resulting in a soluble functional fluorescent imaging tag [71]. CD-TPEs were additionally used to include Dox within the CD cavities. Biocompatible NPs were successfully tested for in vitro cell imaging and displayed significant anticancer activity. Although under experimental in vitro conditions the biological activity was limited by NP internalization and Dox release and was inferior to that of free Dox, CD-TPE were more efficient in tumor suppression in vivo.

Another class of CD-based self-aggregating nanosystems discussed in this review is based on polyrotaxanes and their derivatives. Recently, water-soluble PEG-αCD PR was prepared by carboxylation with succinic anhydride and was endcapped with fluorescent aminophenyl-triphenylporfirin. It was loaded with cisplatin by coordination bonds. Small nanospheres (1.7 nm) were readily taken up by cancer cells. Although increased accumulation was observed in the liver, the formulation preferentially accumulated in the tumor (for at least 168 h) and exerted improved antitumor activity when compared with the free drug [72].

Water solubility can also be achieved by decorating PR CDs with other polymers. In the case of PEG_4.2k_-αCD endcapped with βCD, d-α-tocopheryl polyethylene glycol 1000 succinate (TPGS) was introduced as a solubilizing component [73]. Additionally, αCD was conjugated with 10-hydroxycamptothecin. Although TEM revealed a small, near-spherical morphology, the sample was prepared in DMSO and, like in the above example, possibly reflects the morphology of a single supramolecule. The aggregation pattern, if any, in water or under physiological conditions was not further explored. Nonetheless, the inherent anticancer activity of TPGS [74] contributed to the in vitro and in vivo activity of the drug delivered by the PR.

The ability of a water-insoluble linear β-1,3-glucan curdlan to form a triple-strand helix during the denaturation–renaturation process was exploited to form drug-loaded micelles [75]. Since the cross-section of a curdlan single strand corresponds well to the size of βCD, the CD was threaded during the denaturation phase to form an amphiphilic polymer. Partial threading results in a hydrophobic character of the remaining chain, leading to micelle formation in aqueous solution. The micelles showed improved tumor penetrability and antitumor activity when compared with free camptothecin.

**Table 1 ijms-25-09516-t001:** Summary of published studies on carriers like CD copolymers and host–guest amphiphiles.

Carrier	Responsiveness	Drug	Size (PDI)/ ζ Potential	DL%	Cell Line/ Animal Model	Remarks	Ref.
**CD copolymers**							
βCD-pCL	pH-sensitive ester bonds	Curcumin	150 nm (0.07) −15.8 mV	20	HeLa in vitro and in vivo (heterotopic xenograft)	Significant in vitro cell growth and in vivo tumor inhibition. Improved cancer cell uptake and tumor accumulation via FA cell targeting	[53]
βCD-PEG-Chol	-	Curcumin	Empty: 147 nm (<0.25) Loaded: 121 nm (<0.25)	62	HepG2 in vitro	Encapsulated curcumin attenuated oxidative stress in vitro. Improved in vitro anticancer activity observed. In LPS-induced inflammation in vitro model, encapsulated curcumin reduced production of pro-inflammatory cytokines (IL-6 and TNα).	[54]
(Dex-SS)_n-_βCD-(PCL)_14_ (FA-Dex-SS)-βCD-(PCL)_14_	GSH-sensitive disulfide (SS) bridge	Doxorubicin and SPIO (diagnostic)	Empty: 66 nm (0.12) Loaded: 127 nm	10 (Dox) 11 (SPIO)	HepG2 in vitro	(FA-Dex-SS)-βCD-(PCL)_14_ demonstrated higher antiproliferative activity than (Dex-SS)_n_-βCD-(PCL)_14_ or micelles without the SS linker. Therapeutic efficiency of micellar Dox lower than free Dox	[56]
βCD-PAMAM 3G	-	NO	-	-	*Escherichia coli* (*E. coli*) and *Staphylococcus aureus* (*S. aureus*) in vitro biofilms. *S. aureus* in vivo model of chronic rhinosinusitis	Dendron incorporation significantly prolonged NO half-life (from 3 s in small molecule prodrugs to 30 min in dendron). Efficient inhibition and degradation of bacterial biofilms in vitro and in vivo after intranasal spray. Reduced inflammation of nasal cavity. No local or systemic adverse effects observed.	[58]
DSPE-mPEG_2k_-FA/NH_2_Pr-CD-C_12_	pH-sensitive	Ginsenoside Rg3 and quercetin (1:1) *	Co-loaded: 110 nm (0.3) 6 mV	12 (Rg3) 6 (Querc)	CT26 in vitro and orthotopic CT26-Luc in vivo	~90% of drugs released after 48 h at acidic pH. Co-loaded formulations stable up to a week. Increased antiproliferative and antimetastatic activity of drug combination. Induced ICD, TME remodeling in vivo.	[62]
NH_2_Pr-CD-C_12_		Melarsoprol	Loaded: 110 nm (0.32) 9 mV	11	Hepa1–6 (murine hepatocellular) in vitro and orthotopic in vivo	TME remodeling in vivo. Enhanced immunotherapy activity of anti-PD-L1. ~90% of drug released after 24 h at acidic pH (50% at pH 7.4). Increased antiproliferative and apoptotic activity of encapsulated drug. Improved tumor accumulation and antitumor activity when compared with FA-free formulation.	[63]
R_7_-CD-Man_14_	ATP-sensitive	Norfloxacin	Empty: 264 nm Loaded: 302 nm	11	*E. coli* in vitro and in vivo model of abdominal sepsis	Complete release of drug within 11 h upon ATP exposure. Better biocompatibility than PEG-based controls. Drug-loaded nanorods exhibit superior antibacterial activity in tested models than empty rods or free drug.	[64]
**Host-guest amphiphiles**							
Sulfonamide-βCD and Mtx-SS-Ad	GSH-sensitive disulfide bridge	Methotrexate	78 nm (0.245) 11.6 mV	14.2	78 nm (0.245) 11.6 mV	Sulfonamide-based targeting improved cancer cell sensitivity towards Mtx. Cytotoxicity toward normal cells similar to CaCo-2 and A549 cancer cells	[65]
[PHEMA-g-(PCL-BM:βCD-*star*-PMAA-*b*-PNIPAM)] (*w*/*w* 1:3)	pH (PMAA)- and thermo (NIPAM)-sensitive	Doxorubicin		9.7	MCF-7 in vitro	Increased efficacy of encapsulated Dox (8.3 vs. 1.75 μg/mL for encapsulated and free Dox, respectively)	[66]
Pc-CD:Fc-PEG	Ad-QRH *	Pc	141 nm (0.134)	-	EGFR-overexpressed HT29 (human colorectal adenocarcinoma) in vitro and in vivo (heterotopic xenograft)	Significantly higher photodynamic activity in EGFR-sensitive cells pretreated with Ad-QRH *. Anticancer activity in cells without pretreatment indicates micelle degradation followed by Pc activation. Intravenous administration followed by intratumoral Ad-QRH * application increased tumor fluorescence intensity 5-fold, and NIR irradiation completely eradicated tumors.	[68]
HSPC/βCD-NHC_12_H_25_:Cpt-SS-CD:Cpt-SS-cRGDfk	GSH-sensitive disulfide bridge	Ce6 (PDT) Cpt-SS-CD	Empty: 24 nm Loaded: Ce6 25 nm Ce6 + Cpt 112 nm (Cpt-SS-CD: Cpt-SS-cRGDfk 90:1) 3.9 mV	Ce6 3.2 Cpt 4.4	U87 in vitro and in vivo (heterotopic xenograft)	GSH-sensitive Cpt release combined with photodynamic therapy resulted in synergistic anticancer activity. Improved biodistribution of cRGDfk-labelled dandelions when compared to free drugs.	[69]
AM_7_CD-HAAd	Enzyme-sensitive	Chlorambucil	Empty: 82 nm (0.21); −40.2 mV Loaded: 188 nm (0.27); −19.4 mV	9.4	A549 in vitro	Increased efficacy of encapsulated chlorambucil. Upon HAse exposure, 90% of drug released within 12 h. Strong ATP binding of AM_7_CD.	[70]
CD-TPE:Dox	pH-sensitive	Doxorubicin	165 nm	67	A549 in vitro and 4T1 in vivo syngeneic breast cancer model	Under mild acidic conditions (pH 5.4), 70% of Dox was released due to Dox protonation and solubility shift. After 24 h exposure in vitro antiproliferative activity lower than free Dox. CD-TPE:Dox was more efficient than free Dox in reducing tumor size (6.27 vs. 4.15 mm^3^).	[71]
**Polyrotaxanes**							
Cur/βCD	-	Camptothecin	Empty: 27 nm Loaded: 32 nm	1.2	Human hepatoblastoma HepG2 in vitro and murine breast 4T1 in vitro and in vivo	Cur-CD PR assembled upon Cur helix renaturation in 28% yield in a CD concentration-dependent manner. PR micelles significantly reduced breast tumors in vivo.	[75]
HA/αCD-PEG_2k_	pH-sensitive	Paclitaxel	Loaded: 395 nm (0.16) −15.2 mV	-	HUVEC human normal endothelial cells (2D in vitro) Human A549 lung cancer cells (2D and 3D in vitro)	Micelles stable over 5 days under physiological conditions (pH 7.4) with pH-dependent drug release (90% at pH and 70% at 7.4 within 52 h). Similar in vitro activity to free drug observed in cancer cells but reduced toxicity in normal cells. Micelles demonstrated better cancer cell spheroid penetrability than free probe.	[76]
PLys(BM)/αCD-PEG_5k_	pH- and redox-sensitive	Chlorin e6	120 nm (0.16) −3 mV	-	Human hepatoma LM3 cells (in vitro and in vivo heterotopic xenograft)	Reductive release and activation of Ce6 and led to production of ROS upon laser stimuli. De-capping of protonated BM under reduced pH (6.5) improved cell uptake. Improved in vitro and in vivo antitumor photodynamic effect when compared to free photosensitizer.	[77]
mPEG_2k_-PLGA_5k_/αCD-It PPR	GSH-sensitive	Oxaliplatin-desmethyl naproxen prodrug	190 nm (0.2) −19.7 mV	-	Murine CT-26 colorectal cancer cells (in vitro and in vivo) HT-29 and HCT-116 human colorectal cancer cells (in vitro)	Improved and cell-dependent in vitro anti-cancer activity when compared with free Oxa or formulation without It. COX2 suppression and GSH sequestration enhanced Oxa efficacy. Unlike Oxa, formulation was well tolerated in animal model. It was accompanied by strong anti-tumor activity.	[52]

Abbreviations: pCL polycaprolactone; FA folic acid; Ad amantadine; HepG2 human hepatic cancer cells; PDMA poly(2-(dimethylamino)ethyl methacrylate; Chol cholesterol; Dox doxorubicin; [PHEMA-g-(PCL-BM:β-CD-star-PMAA-b-PNIPAM)] poly(2-hydroxyethyl methacrylate)-graft-[polycaprolactone-benzimidazole:β-cyclodextrin-star-poly(methacrylic acid)-block-poly(N-isopropylacrylamide)]; AM hexylimidazolium; HA hyaluronic acid; NH_2_Pr-CD-C_12_ heptakis [2-O-(N-(3″-aminopropyl)-1′H-triazole-4′-yl-methyl)-6-dodecylthio]-β-CD; DSPE 1,2-distearoyl-sn-glycero-3-phosphoethanolamine; HSPC hydrogenated soybean phosphotidylcholine; Ce6 chlorin e6; Cpt-SS-CD camptothecin produg; Pc phthalocyanine-based photosensitizer; Fc ferrocene; QRH * targeting peptide ligand; R rhodamine; Man mannose; OLA oleylamine; Cur curdlan; PR polyrptaxane; BM benzimidazole; It itaconate, PPR poly(pseudo)rotaxane. * molar ratio.

When conjugated with hydrophilic HA, βCD-capped αCD-PEG_2k_ PR chains formed the hydrophobic portion of the amphiphile [76]. The terminal βCD stopper further contributed to the encapsulation of the hydrophobic paclitaxel (~200-fold increase in solubility) while HA promoted selective uptake by cancer cells and led to increased anticancer activity of the drug. The same approach was applied to the construction of micelles composed of αCD-PEG_5k_ PR grafted with poly(L-lysine) derivatized with benzimidazole moieties (PLys(BM)) [77]. Photosensitizer Ce6 was conjugated with αCD via disulfide linkers, and hydrophobic BMs formed an inclusion complex with 2-hydroxypropyl-βCD to form a hydrophilic shell. Under tumoral physiological conditions (pH~6.8), decomposition of the pH-sensitive BM-βCD complex exposed positively-charged BMs and promoted cell uptake. Finally, GSH-rich intracellular environment led to the release of the active molecule which was successfully tested for photodynamic therapy in a subcutaneous and orthotopic animal model of hepatoma.

Similar concept was applied in mPEG-PLGA PPR for anti-inflammatory oxaliplatin (IV) (Oxa) prodrug delivery [52]. Release of inflammatory factors triggered by Oxa can significantly decrease its therapeutic activity. To counter this effect, an anti-inflammatory prodrug was designed by complexing Oxa with anti-COX2 desmethy naproxe (DN). High intracellular GSH concentrations could release DN and deactivate Oxa at the same time. To prevent Oxa loss, αCD was decorated with an itaconic moiety in order to trap GSH via double bond-sulfhydryl addition (Figure 5). Cumulative effects of functional molecules enhanced the therapeutic effectiveness of Oxa.

## 4. Peptide and Protein Delivery

Hydrophilic therapeutic molecules like proteins with intracellular targets hold great potential due to their high specificity towards targeted motifs. On the contrary, conventional chemotherapy can be tailored only for a fraction of potential intracellular protein targets [78]. The rest of the ‘undruggable’ intracellular cancer antigens could be reached by proteins with various inhibitory or enzymatic activities. However, size and hydrophilicity prevent cell membrane crossing, and various strategies for intracellular protein delivery have been devised with varying impact on their stability, functionality, and intracellular colocalization. Encapsulation within polymeric nanovehicles has become a widely used strategy to improve stability and pharmacokinetic profile and to reduce immunogenicity and nonspecific interactions of these biotherapeutics. Nanocarrier toxicity, loading capacity, and spatiotemporal release of functional proteins must be resolved for successful clinical translation.

The most used strategies to encapsulate proteins are based on weak non-covalent interactions which can result in a loss of cargo. An alternative approach to protein delivery based on strong covalent conjugation of proteins with polymers has emerged as a preferable option. It balances the benefits of protective polymer coating and preserved protein functionality by using dynamic covalent bonds based on self-immolative linkers sensitive to various intra- and extracellular stimuli. Moreover, tuning polymer efficacy for enhanced protein loading could be achieved by the addition of structural motifs for complementary noncovalent interactions [79]. However, since therapeutic proteins are characterized by different sizes, shapes, and isoelectric points, the design of the universal protein carriers could be a challenging task. The application of CDs in peptide and protein delivery is illustrated in Table 2.

In that context, CD has been used as a linker for functionalized protein nanoformulations. Conjugation with cell-penetrating peptides (CPP) became one of the most explored approaches to intracellular protein delivery. However, it is often associated with lengthy genetic manipulation which can impact on CPP functionality. Alternatives include chemical conjugation and noncovalent interactions of proteins with CPPs or encapsulation by CPP-decorated vehicles. For example, Kitagishi et al. described the applicability of a simple derivatization to a variety of proteins with different physicochemical properties (green-fluorescent protein_27k_, β-galactosidase_65k_, IgG_151k_). Introduction of adamantane moiety to a protein scaffold enabled decoration with CPP octaarginine (R8-CD^OH^) via Ad:CD inclusion complex (Figure 6) [80].

Derivatization at the optimized R8-CD^OH^ density for maximized cell uptake did not interfere with the structure or the function of the proteins. 

Unlike covalent protein modification by CD complexes, nanovehicles based on selfassembling amphiphilic CD polymers are being developed as well. Recently, Kudruk et al. described biocompatible smart nanocontainers for intracellular delivery of proteins [81]. Bilayer polymersomes were prepared by thin film hydration and subsequent extrusion of amphiphilic hydroxyethylated 6-alkylthio-βCD and were further functionalized by the introduction of a peptide hydrophilic shell of controllable density via CD inclusion of Ad-peptides. Two types of peptide shells were tested. Thio-containing short peptides (*n* = 6–9) yielded a GSH-responsive crosslinked shell by oxidation (SPSVSS), while a thicker polypeptide shell was obtained by introducing polyGlu (*n* = 100 ± 5). The thicker shells were crosslinked by cystamine (PPSVSS). Although some background leakage of the model cargo remained, the crosslinking and denser polypeptide shell in PPSVSS improved the stability of polymersomes and prevented cargo leakage when compared to SPSVSS. PPSVSS were readily taken up by endocytosis and were degraded by the reductive and proteolytic endosomal environment. Negatively charged pGlu also facilitated cargo colocalization within the cytoplasm by affecting the membrane potential of endosomes and destabilizing them. While the pGlu shell remains trapped in the endosome, the cargo leaves the compartment by passive diffusion. Efficient cytosolic delivery was also observed for peptides and oligopeptides with anticancer activity.

Therapeutic vaccination has attracted great interest in the treatment of cancer and viral diseases. The challenge posed by the lack of efficient immunogenic tumor-associated antigens in vivo could be overcome by nanoformulations which could prolong their stability and exposure to the immune system and could also be used for co-delivery with other immunomodulators. They could also improve their delivery to the antigen-presenting cells (APC) and improve the activation of antigen-specific cytotoxic CD8+ T cells (CTL). To exploit the potential of CD-based nanostructures to correctly present antigens to the APCs, Geisshusler et al. evaluated hydrophobic CDs substituted on the primary face with C8 and C12 chains (βCD-C8_7_ and βCD-C12_7_) by thioalkylation [82]. Antigen loading was accomplished by insertion of hydrophobic domains into CD cavity. SFL (SIINFEKL) antigen and its hydrophobic derivative SFL-c (L_4_SIINFEKLA_3_) were internalized by energy-dependent endocytosis and colocalized within lysosomes. Both peptides were adequately processed by APC dendritic cells and presented to T cells in major histocompatibility complex I (MHC I) to activate CD8+/44+ T cells in vitro. Nonetheless, only βCD-C8_7_ was tested in vivo due to the size restrictions required for direct diffusion into the local lymph nodes. The stronger in vivo response it displayed to encapsulated SFL-c indicated this formulation’s potential future application in the delivery of hydrophobic antigens. Stronger response to capped antigens could be explained with higher loading parameters related to insertion of a more hydrophobic species.

**Table 2 ijms-25-09516-t002:** Summary of studies on CDs for protein delivery.

Carrier	Responsiveness	Protein	Size (PDI)/ ζ Potential	DL%	Cell Line/Animal Model	Remarks	Ref.
SPSV_SS_ PPSV_SS_	Enzyme- and GSH-sensitive	Phalloidin (heptapeptide; staining) α-amanitin (octapeptide; anticancer)	~125 nm/ −14 to −7 mV 145 nm/ −15 mV	-	HeLa (cancer) and HUVEC (endothelial) in vitro	Background cargo leakage observed for the model dye cargo but was not investigated for the biological cargo. Bioactive peptide cargo colocalized into the cytoplasm and inhibited cell proliferation (cancer and endothelial).	[81]
βCD-C8 βCD-C12	-	SFL; SFL-c	C8-SFL: 178 nm (0.08) C12-SFL: 129 nm (0.1) C8-SFLc: 358 nm (0.22); C12-SFLc:: 287 nm (0.17) (*w*/*w* βCD/peptide 1:10) ~−35 mV all	-	IC-22 macrophages and BMDC in vitro	Encapsulated peptides induced DC maturation and were presented on surface MHC I. Lower DC maturation by SFL-c might indicate the need for intracellular decapping for MHC I presentation. In vivo stimulation with βCD-C8_7_:SFL-c induced a faster and stronger response of CD8^+^ T cells than βCD-C8_7_:SFL or control poly(I:C)-adjuvanted SFL.	[82]
βCD-(DIBO-Lys)_7_	-	BSA DNase I Nrf2	~150 nm - 80 nm/−15 mV	14 20 -	HeLa in vitro (DNase I) Hepatocytes in vitro (Nrf2) APAP-induced in vivo hepatic injury murine model	βCD-(DIBO-Lys)_7_ colocalized proteins in nucleus. Therapeutic DNase I caused HeLa cell apoptosis. Therapeutic Nrf2 triggered antioxidative response in hepatocytes exposed to H_2_O_2_ and in in vivo model of hepatic toxicity after intravenous administration.	[83]
βCD-(DIBO- Lys)_7_		Sod Cat	156 nm (βCD-(DIBO-Lys)7/Sod/Cat *w*/*w*/*w* 8:1:1) 3.5 mV	-	RAW264.7 in vitro DSS-induced in vivo murine colitis model	Synergistic anti-inflammatory effect of encapsulated Sod and Cat on secretion of pro-inflammatory factors. After oral administration Sod/Cat/βCD-(DIBO-Lys)_7_ accumulated in inflamed colon and attenuated colitis symptoms.	[84]

Abbreviations: SPSVSS hydroxyethylated 6-alkylthio-βCD:Ad-short peptide; PPSV_SS_ hydroxyethylated 6-alkylthio-βCD:Ad-pGlu; DIBO dibenzocyclooctyne; BSA bovine serum albumin; Nrf2 nuclear factor E2-related factor; APAP acetaminophen; Sod superoxide dismutase; Cat catalase; DSS dextran sulphate sodium; SFL MHC I restricted ovalbumin-derived model peptide SIINFEKL; SFL-c caped SFL L_4_SIINFEKLA_3_; BMDC bone marrow dendritic cells; MHC I major histocompatibility complex I.

Cationic polymers are often used for the formation of polyion complex-like nano-aggregates with proteins at the core. When hydrophobic tethers are introduced, electrostatic interactions with negatively charged amino acid residues are complemented by the formation of hydrophobic interactions with hydrophobic domains of the protein, as demonstrated by Zeng et al. [83]. Dibenzocyclooctyne (DIBO) was used as a hydrophobic linker to graft βCD with cationic Lys (βCD-(DIBO-Lys)_7_) (Figure 7). This wind chime-like lysine modified CD for nuclear targeting was used to complex a variety of proteins (bovine serum albumin BSA, DNase I, nuclear factor E2-related factor Nrf2) [83]. BSA/βCD-(DIBO-Lys)_7_ was internalized by active transport and colocalized in the nucleus as a part of the CD complex. The nature of CD cationic moieties was found to have influence on cellular distribution and Arg-substituted CDs dispersed BSA throughout the cytoplasm. The exposed surface of CDs was used to decorate aggregates with hepatocyte-targeting galactose ligands (Ad-Gal).

βCD-(DIBO-Lys)_7_ was also used for co-delivery of anti-inflammatory superoxide dismutase (Sod) and catalase (Cat) by oral administration for the treatment of inflammatory bowel disease [84]. The simple encapsulation process is highly repeatable and simple. The applicability of this system to various types of proteins and administration modes make it promising for future intracellular delivery of therapeutic proteins. Additionally, βCD-(DIBO-Lys)_7_ can be used to encapsulate proteins by complementary mechanisms. In the case of multi-protein self-assembly (RNaseA/DNaseI), the RNase component was decorated with a self-immolative Ad linker and derivatized with the βCD-(DIBO-Lys)_7_ [85]. This cationic complex was actively internalized by several endocytotic mechanisms (e.g., caveolin, lipid raft), escaped the endosomal compartment to colocalize in the nuclei, and exerted cytotoxic activity in vitro and in vivo.

## 5. Nucleic Acid Delivery

The development of gene-based therapies is closely related to the design of new vehicles for DNA delivery able to surpass issues related to low stability, unsatisfactory bioavailability and biodistribution, immunogenicity, and insufficient cell specificity. Discovery of RNA-based interference opened new avenues for modulation of gene and protein activity by post-transcriptional gene silencing by short oligonucleotides (19–25 base pairs). More recently, new non-coding RNA species with potential therapeutic application have been discovered [86]. Nevertheless, some common characteristics like limited complex stability, enzymatic cargo instability, and limited cellular uptake impair their clinical applicability. Despite the progress already achieved in the field of NA delivery, the search for efficient vehicles is still ongoing [87,88]. Most of the nanovehicles developed so far are based on NA complexation via electrostatic interactions with negatively charged NAs. Among them, those composed of amphiphilic molecules often present compartmentalized lyophilic and hydrophilic domains able to bind not only NAs but also hydrophobic chemotherapeutics or adjuvants. It provides the opportunity to aim at various complementary targets and improve therapeutic outcomes. Nonetheless, different physicochemical properties (size, shape, charge) of NA therapeutics can demand individual approaches to design the appropriate carrier for each NA species.

Advantages of CDs such as biocompatibility and biodegradability, high transfection efficacy, and easy functionalization have been applied for delivery of NAs and resulted in CALAA-01, the first polymer-based silencing siRNA carrier to enter clinical trials. However, dose-limiting toxicity related to carrier stability resulted in withdrawal of that particular formulation [89,90]. Recently, progress and the state-of-the-art in the application of CD-based nanocarriers for NA delivery has been compiled in several specialized reviews [91,92]. The overview of the latest examples described in this review is presented in Table 3. Similarly to protein delivery, cationic derivatives of CDs are widely applied for NA delivery as well.

Adaptations which occur in the TME during tumorigenesis or are triggered by anticancer therapies induce gradual shifts towards immunosuppressive phenotypes by recruitment and reprogramming of immunosuppressive cells like tumor-associated macrophages M2, regulatory T cells, and myeloid-derived suppressor cells and can contribute to the immunotherapy resistance that is observed in the majority of patients. To address the limitations of existing immunotherapy, the next generation of immunotherapy that modulates the function of various tumor-associated immune cell populations can be used to enhance the antitumor immune response. Downregulation of colony-stimulating factor-1 receptor (CSF-1R) can reprogram immunosuppressive M2 tumor-associated macrophages into pro-inflammatory anticancer phenotype M1. NH_2_Pr-CD-C_12_ has emerged as a model amphiphilic CD system for delivery of hydrophilic biomolecules and is increasingly used in structure–activity studies. In a study by Sun et al., it was functionalized by M2-targeting ligand M2pep and protective PEG by DSPE-mPEG_2k_ insertion [93]. The formed polyplex provided sufficient siRNA stability (24 h) for efficient M2 reprograming in vitro and associated cancer cell apoptosis. Complexed siRNA avoided the fast degradation observed for naked NA after systemic administration and preferentially accumulated in tumors (2.4-fold vs. nontargeted polyplex), reducing its weight and size by 70%. Simultaneously, elevated infiltration of CD8+ T cells and remodeling of tumor associated macrophages (TAM) occurred in an immunocompetent animal model (Figure 8). These results are comparable to some other systems functionalized with the same targeted ligand [93].

DSPE-mPEG_2k_ postinsertion is a commonly used strategy to derivatize nanoassemblies of amphiphilic CDs. However, mono fatty acid-based insertion ligands can also be used for this purpose. Stearic acid-PEG-COOH was esterified by sialic acid, an M2 macrophage targeting ligand with little expression in other macrophage populations [94].

Cationic polymers are usually efficient in complexing and stabilizing short chain RNA, but the excess of positive charge is associated with the observed side-effects. Therefore, introduction of a helping, negatively charged polymer could improve electrostatic interactions within the polyplex, reduce the charge, and improve stability under physiological conditions. Anionic CD was prepared by a sequential esterification of the secondary face and sulphation at position 6 (C_12_-CD-SO_3_) [95]. Optimization of composition was needed to avoid displacement of therapeutic load (*w*/*w* NH_2_Pr-CD-C_12_/siRNA/C_12_-CD-SO_3_ 8.5:1:5.3). Both coated and non-coated polyplexes were equally efficient in siRNA stabilization and protection. Both formulations had similar intracellular residence time, but the endo-lysosomal escape occurred earlier for the non-capped CD (12 vs. 24 h). Despite that, gene silencing remained similar over the 72 h exposure.

Compartmentalized domains of cationic amphiphilic CDs can be exploited for co-delivery of hydrophobic drugs and therapeutic NAs and have previously been exploited, especially in anticancer therapy [92,96,97]. NH_2_Pr-CD-C_12_ was exploited for co-delivery of docetaxel (Dtx) and anti-NF-κB siRNA [98]. The authors hypothesized that downgraded activity of NF-κB, which is heavily involved in chemoresistance, might enhance Dtx activity. A drug-loaded CD was obtained by hydration and sonication of a thin film and was functionalized with FA and PEG by postinsertion. Comparison of Dtx loading with that observed for CD inclusion complexes revealed that improved drug loading was achieved by the amphiphilic CD because of the increased number of binding compartments. The insertion of functional lipids resulted in the increased size of CDplexes but did not influence drug loading parameters. Morphological analysis of polyplexes by TEM revealed spherical structures but without any finer insight into the structural features or the position of incorporated therapeutic molecules. More detail could be obtained by techniques like small-angle X-ray scattering (SAXS). The analysis of scattering profiles of heptakis [2-(ω-amino-oligo(ethylene glycol))-6-deoxy-6-hexadecylthio]-β-cyclodextrin and heptakis [2-(ω-amino-oligo(ethylene glycol))-6-deoxy-6-dodecylthio]-β-cyclodextrin (SC16NH_2_) complexes with ctDNA and pDNA revealed that the unilamellar vesicles formed by the CD undergo rearrangement upon interactions with ctDNA, and that the short segments of double helix ctDNA become sandwiched between the positively charged layers of multilamellar vesicle [99]. Still, the morphology of CDplexes depends on the CD:DNA ratio. Unlike the counterpart with longer hydrophobic chains (C16) considered above, CD substituted with shorted C12 chains forms micelles, and ctDNA interacts electrostatically with the surface of micellar clusters which are stabilized by the ctDNA backbone. Correlating morphological characteristics of CDplexes with transfection efficacy can be difficult since CDplexes are equilibrium systems. Additionally, kinetic factors should also be considered. Nonetheless, higher transfection potency was observed for the multilamellar CD-C16 polyplex. Although SC16NH_2_ transfects pDNA as well, under the experimental conditions used by the authors, there was no evidence for complexation with pDNA provided by SAXS other than the evidence of electrostatic interactions. In another study based on CD-C16 polyplexes, both SAXS and cryo TEM revealed that under certain CD-pDNA ratios, the resulting CDplexes also assume multilamellar morphologies [100]. Additionally, branched polycationic chains produced more stable CDplexes than linear poly or mono cationic chains.

The nature of cationic domain also determines the composition of protein corona that forms when CDplexes are incubated in human plasma (HP). Coagulation, complement, and lipoproteins were the most abundant and could guide the fate of the CDplexes, directing blood clearance, stability, crossing of biological barriers, and biodistribution. Transfection was found to be cell- and CD-dependent and was also modulated by the presence of HP. Although protein corona can be beneficial for interactions with cell receptors, it is generally considered that it obstructs interactions with the endosomal membrane and impair endosomal escape. In the case of drug-siRNA-loaded CDplexes, PEGylation of NH_2_Pr-CD-C12 successfully prevented protein corona formation in the presence of HP, as evidenced in preserved size. However, biological repercussions of protein corona absence were not pursued in this case. Nonetheless, doubly loaded CD/PEG/FA exhibited synergistic anticancer effects by efficient intracellular release of therapeutic cargo both in vitro and in vivo.

Similar cationic CD substituted with tetraethyleneimine on the primary rim and haxanoyl tails on the secondary (TEI_7_-CD-C16_14_) was used to deliver anti-MAPK and/or Rheb siRNA, and complement Dtx activity in prostate cancer cells [101]. In this case, reduction of size was observed upon formation of lamellar CDplex. Knockdown of one or both pathways implicated in cancer cell proliferation and survival boosted cytotoxic Dtx activity in cell-dependent manner (LNCaP > PC3).

**Table 3 ijms-25-09516-t003:** Summary of studies focused on CDs for nucleic acid delivery.

Carrier	Functionalization	NA	Size (PDI)/ ζ Potential	Cell Line/Animal Model	Remarks	Ref.
NH_2_Pr-CD-C12/DSPE-mPEG_2k_PEG/DSPE-mPEG_2k_M2pep	M2pep	*CSF-1R* siRNA	252 nm (0.25) 10.8 mV (*w*/*w* NH_2_Pr-CD-C12/siRNA 10:1)	Human THP1 and RAW 264.7 murine monocytes Human PC-3 and murine TRAMP-C1 prostate cancer cells in vitro and in vivo (heterotopic xenograft)	Complexed siRNA stable up to 24 h in 50% serum. M2pep significantly increased M2 uptake, reduction of *CSF-1R* mRNA (~50%), and reprogramming to M1 (~50%). Production of M1 factors was accompanied by cancer cell apoptosis. Increased tumor accumulation was observed 12 h after administration (3.5-fold) when compared with non-targeted formulation. Reduction of tumor was accompanied by immune remodeling of TME, and redistribution of immune cells and factors was observed.	[93]
NH_2_Pr-CD-C12/ DSPE-mPEG_2k_PEG/ C18-PEG-SA	SA	*CSF-1R* siRNA	246 nm 29 mV	Human THP1 and RAW 264.7 murine monocytes Human PC-3 and murine TRAMP-C1 prostate cancer cells in vitro	Insertion of PEG and SA had no impact on siRNA complexation. PEGylation prevented CDplex aggregation under physiological conditions. SA ligand improved macrophage uptake and reduced targeted mRNA expression, resulting in reprograming M2 (50%). M1 cytokines increased apoptosis of prostate cancer cells.	[94]
NH_2_Pr-CD-C12/ C12-CD-SO_3_	-	*KAT2* siRNA	161 nm (0.16) and 24 mV (vs. 164 nm (0.46) and 34 mV for NH_2_Pr-CD-C12)	Human HL-60 myeloid leukemia in vitro	Coated polyplex possesses bilayer structure with hydrophobic C12 interactions contributing to polyplex uniformity. Maximal internalization at 6 h (decreased by half at 24 h)	[95]
TEI_7_-CD-C16_14_	-	p*42*-MAPK and Rheb siRNA	170 nm/58 mV 100 nm/25 mV (CDplex)	Human LNCaP and PC3 prostate cancer cells (in vitro). Human U87 MG and rat C6 glioblastoma cells, GL-261 mouse glioma cells (in vitro)	Transfection and mRNA knockdown observed in various cell types.	[101]
Spermidine-CD:Ad-PVA-PEG	pH sensitive acetal linker	siRNA	130–260 nm (0.08–0.18) −23–−61 mV	A549 in vitro	Formation of polyplex followed by hydrophilic polymer insertion resulted in more stable CDplexes, while the length of PEG-directed size and ζ. siRNA complexation led to size contraction.	[34]
**Drug-NA co-delivery**						
NH_2_Pr-CD-C12 NH_2_Pr-CD-C12/PEG NH_2_Pr-CD-C12/PEG/FA	FA	Docetaxel *RelA* siRNA	100 nm (0.27)/10 mV 122 nm (0.24)/40 mV (PEG) 125 nm (0.26)/9 Mv (PEG/FA)	Murine CT26 CRC line	CD derivatization significantly increased the size of loaded CDplex. pH-dependent drug release observed. Synergistic anticancer activity was observed and was especially pronounced for targeting CDplex.	[98]
CatCD:Fc-prodrug	H_2_O_2_	*MTH1* siRNA Ferrocene	80 nm (polydisperse) 4.9 mV	HeLa in vitro MDA-MB-23 3D speroid (in vitro)	Oxidation products of Fc-prodrug include ROS and p-quinone methide (GSH scavenger). NPs are taken up by cholesterol-dependent endocytosis and are colocalized within cytoplasm. Fe-siRNA co-delivery improves Fe chemodynamic activity in 2D and 3D in vitro models.	[102]
βCD-PCL-Ad:βCD-PCL-PDMAEMA	-	Doxorubicin *Nur77DDBD pDNA*	~200 nm ~20 mV	Human HepG2/MDR1-Bcl2 hepatoma cells (in vitro)	Complexation improves drug loading and release and is efficient in transfecting MDR cancer cells. Suppression of gene related to drug resistance improved Dox activity.	[103]

Abbreviations: NH_2_Pr-CD-C_12_ heptakis [2-O-(N-(3′′-aminopropyl)-1′H-triazole-4′-yl-methyl)-6-dodecylthio]-β-CD; TEI tetraelthyleneimine; FA folic acid; CRC colorectal cancer; SA sialic acid; Cat cation.

Seripracharat et al. prepared CD polyplexes crosslinked by grafted polyvinyl alcohol-g-PEG (PVA-g-PEG) via an Ad-CD inclusion complex [34]. Ad was introduced via a pH acetal group with PVA hydroxy groups (Ad-PVA-PEG2k and Ad-PVA-PEG5k) for facilitated siRNA release under acidic physiological conditions. Detailed studies of complexation order (polyplex-first or Ad-CD crosslinking first) and PEG/cation (putrescine-spermidine-spermine) length were conducted. The order of complexation and the length of PEG had decisive influence on the physical properties of CDplexes, and more stable CDplexes were obtained by addition of Ad-PVA-PEG on previously formed polyplexes. Additionally, spermidine exhibited the most efficient NA binding.

Host–guest interactions were used by Raj et al. for co-delivery of ferrocene prodrug of transitional metal for chemodynamic therapy in combination with siRNA for silencing of oxidative damage-repairing protein MTH1 [102]. The formation of an amphiphilic 1:1 complex led to the formation of spherical nanoparticles with exposed positively charged faces.

The nanocomplex of βCD-g-(poly(ε-caprolactone)-adamantyl (βCD-PCL-Ad) and βCD-g-(poly(ε-caprolactone)-poly(2-(dimethylamino) ethyl methacrylate) (βCD-PCL-PDMAEMA) formed by Ad-CD inclusion was used for drug/pDNA co-delivery [103]. The complex with layered domains was more efficient in Dox encapsulation than βCD-PCL-Ad alone due to the existence of hydrophobic CD cavity and PCL layers. Combination of Dox and pro-apoptotic pDNA led to improved Dox activity in the MDR cancer cells.

## 6. Final Remarks and Future Perspectives

Over time, CDs have been singled out as attractive biocompatible structural elements of nanovehicles for delivery of therapeutic molecules with wide range of structural and physicochemical characteristics. Availability of two faces with a high number of modifiable sites can be exploited for design of derivatives with adjustable ligand densities. Advanced methods that can be applied in “core-first” or “ligand-first” methodology for CD modification give rise to modulation of polymer chain characteristics which can be combined with selective face derivatization. Judicial choice of polymer composition and density can lead to self-assembly and formation of diverse morphologies, ranging from single-molecule and classical micelles to multilayered vesicles. Nonetheless, despite the development of synthetic methods related to selective CD activation and synthesis of polymers with well-defined composition, some challenges still need to be resolved. As synthesis often involves multiple steps, production scale-up is often problematic. Also, issues related to synthetic reproducibility are often not addressed in the pre-clinical reports presented in this overview of recent development in the field.

The adjustable character of CDs derivatized with polymers can be complemented by the inherent CD cavity, which adds another layer to the synthetic diversity of CDs. It can be used for host–guest inclusion, leading to additional functionalization of CDs which can simplify the synthetic procedure. Also, it presents an additional reservoir for drug incorporation and complements the hydrophobic layer provided by grafted polymers. Furthermore, as observed in proteins, it can participate in interactions with biological macromolecules that contain hydrophobic domains. The ability to form inclusion complexes and establish electrostatic interaction with negatively charged therapeutic load (nucleic acids and some proteins/peptides) can facilitate the design of universal nanovehicles, and some CD derivatives (e.g., NH_2_Pr-CD-C12) have been used for efficient delivery of hydrophobic drugs and hydrophilic biomolecules. In nucleic acid delivery, transfection efficacy is often comparable or even superior to used controls (PEI_25k_ and transfectamines).

Compared to amphiphilic CD nanovehicles, poly and poly(pseudo)rotaxanes are still not as represented and explored, especially from a self-assembling point of view. The low efficiency of PR synthesis can limit their preparation due to concomitant dethreading during the capping reaction. Additionally, the control of the threading ration is another parameter crucial for wider applicability of this class of CD derivatives.

Taken from the examples presented in this article, polymer-based self-assembling CD nanostructures have favorable properties to increase the solubility and improve the stability of the load and thereby influence their overall pharmacokinetic and dynamic profiles. Improved in vitro and in vivo therapeutic outcomes have been observed in most of the cases presented. Nonetheless, lack of detailed understanding of structure–activity relationship and its optimization limit the insight into a true potential of this class of CD derivatives. Nonetheless, as evidenced by the recent examples, modular amphiphilic CDs hold great potential for future development and application as nanovehicles of therapeutic load.

## Figures and Tables

**Figure 1 ijms-25-09516-f001:**
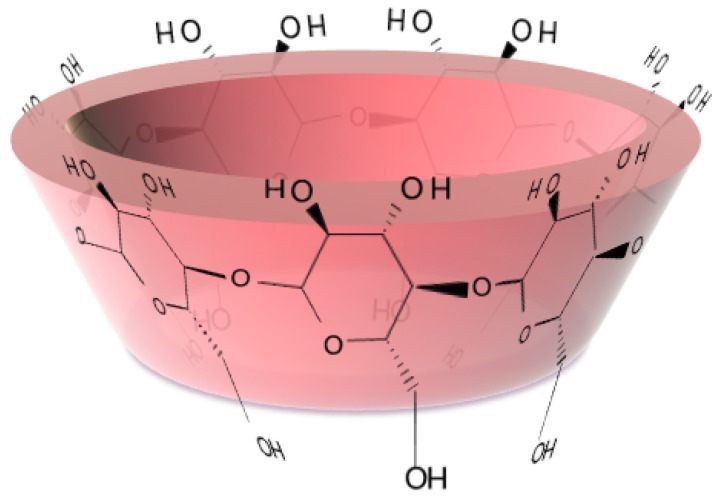
Representation of βCD.

**Figure 2 ijms-25-09516-f002:**
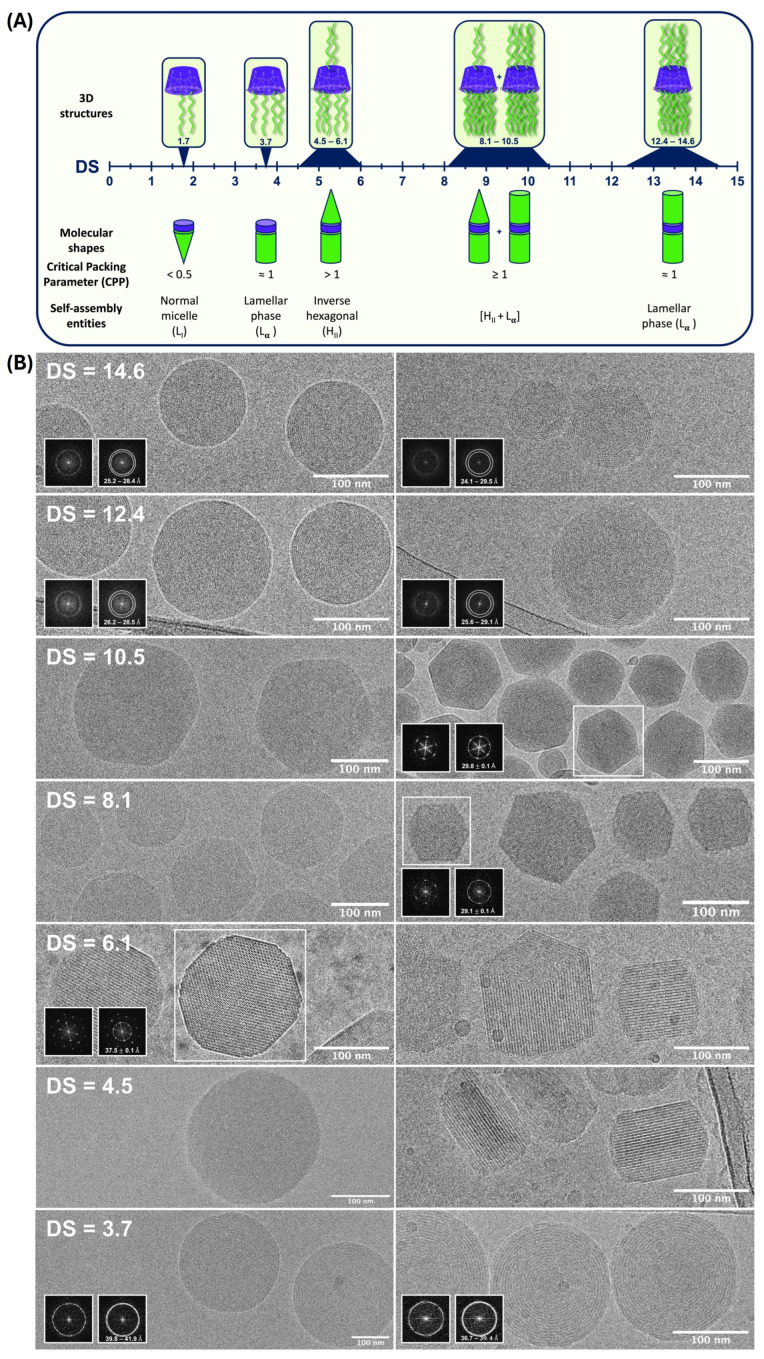
Influence of structural parameters on nanoassembly morphology illustrated by example of acylated cyclodextrins (β-CD-C10): (**A**) influence of critical packing parameters on morphology; (**B**) cryo TEM images of non-PEGylated (**left**) and PEGylated (**right**) β-CD-C10. Reproduced with permission from [21].

**Figure 3 ijms-25-09516-f003:**
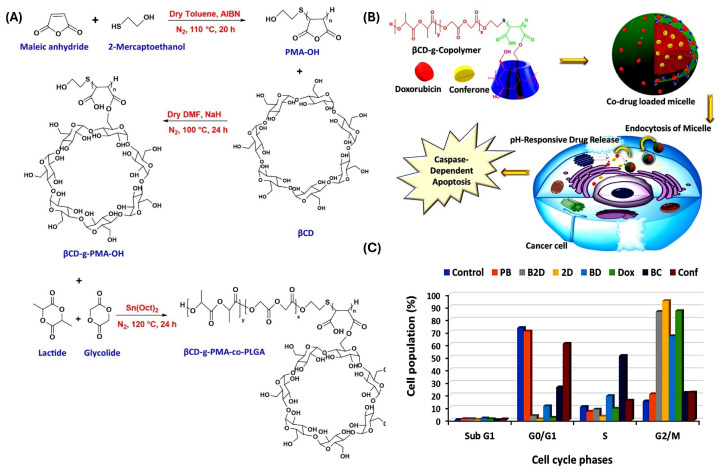
Amphiphilic βCD-g-PMA-co-PLGA for co-delivery of drug and adjuvant. (**A**) Schematic representation of polymer synthetic sequence; (**B**) schematic representation of drug/adjuvant activity and delivery; (**C**) quantitative diagram of therapeutic outcome (cell population distribution) after treatment. PB blank micelles; B2D coloaded micelles; BD and BC singly loaded micelles; Dox and Con free drugs; 2D free drug combination. Reproduced with permission from [57].

**Figure 4 ijms-25-09516-f004:**
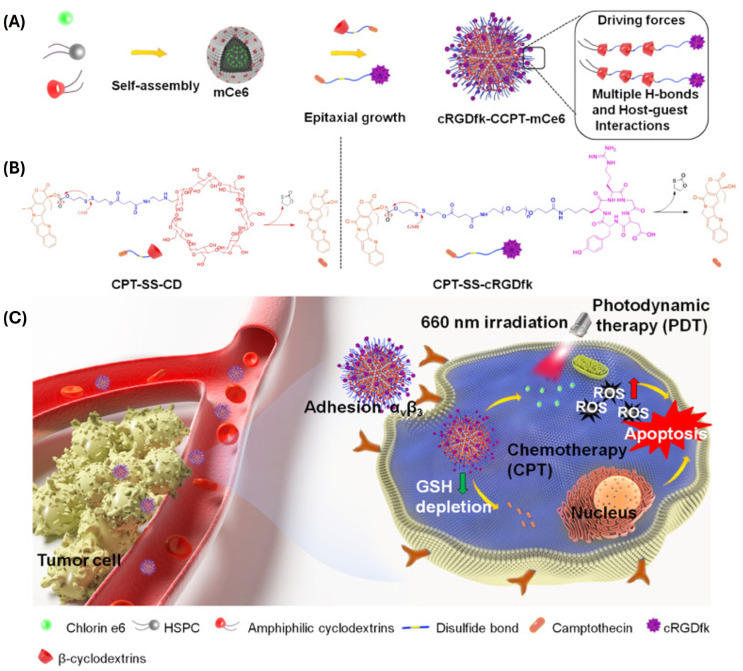
Schematic representation of dandelion-like CD-based micelles for dual chemo and phototherapy: (**A**) βCD-NHC_12_H_25_/HSPC self-assembly and dandelion shell growth; (**B**) stimuli-sensitive degradation of camptothecin-CD/camtothecin-cRGDfk prodrugs and drug release; (**C**) active targeting of cells with high integrin αvβ3 expression is followed by GSH-induced vesicle degradation and release of active ingredients. Photodynamic therapy complements chemotherapeutic activity upon irradiation. Reproduced with permission from [69].

**Figure 5 ijms-25-09516-f005:**
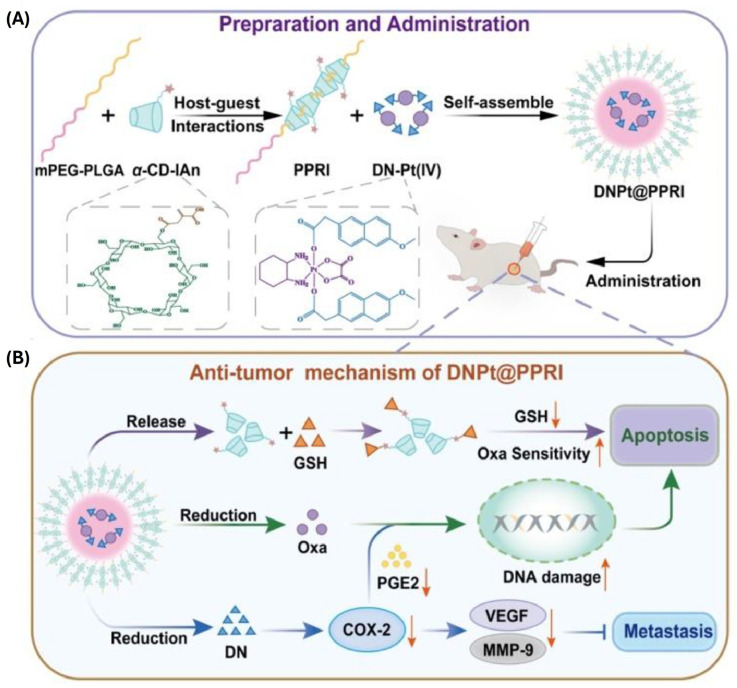
Schematic representation of multifunctional mPEG-PLGA poly(pseudo)rotaxane for delivery of Oxa/anti-inflammatory prodrug: (**A**) PPR self-assembly; (**B**) stimuli-sensitive release of Oxa and DN, and GSH trapping by αCD-itaconate. Reproduced with permission from [52].

**Figure 6 ijms-25-09516-f006:**
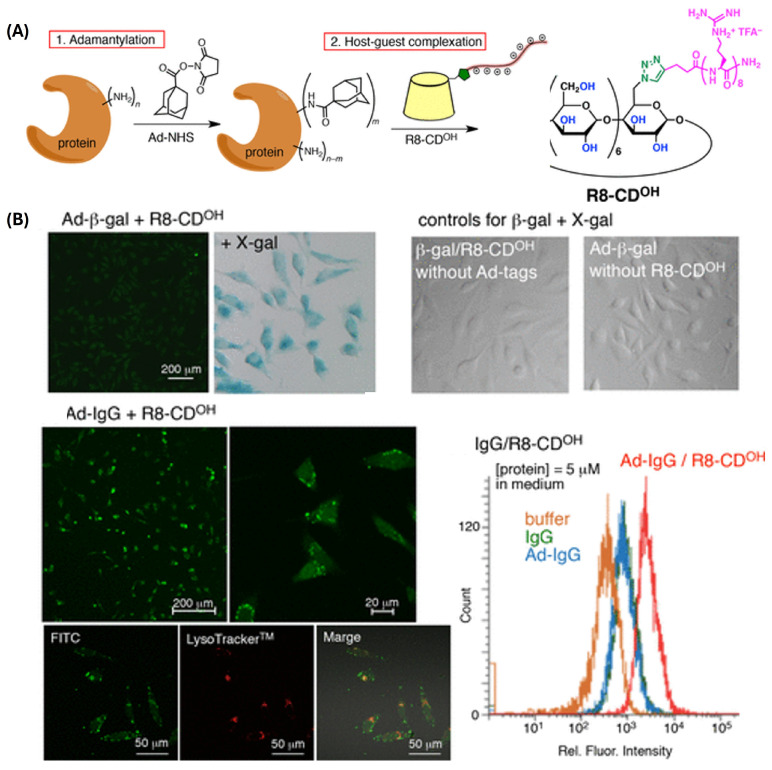
Self-assembled CDs for protein delivery: (**A**) scheme of a simple synthetic workflow for protein derivatization and functionalization via host–guest insertion. The structure of derivatized CD is presented; (**B**) confocal micrographs and flow cytometry histograms of cellular uptake of proteins (β-galactosidase and IgG) modified with Ad-NHS and R8-CDOH. Reproduced with permission from [80]. Copyright 2020 American Chemical Society.

**Figure 7 ijms-25-09516-f007:**
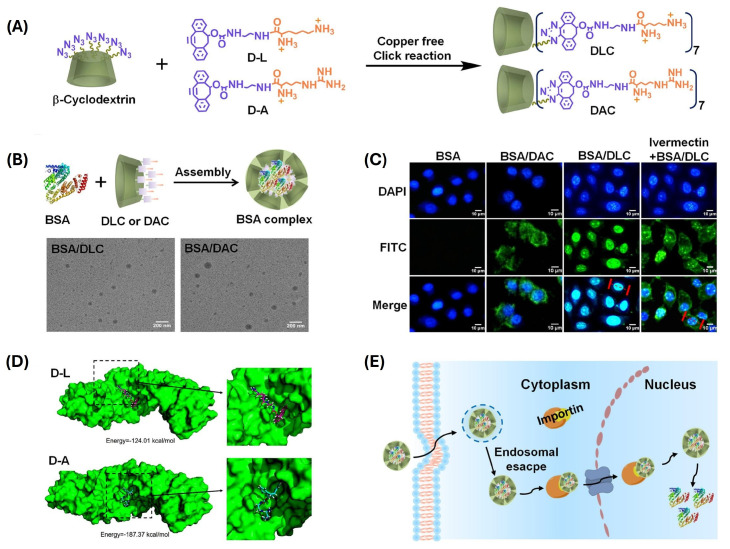
Polyion-based CD nano-assembly for intracellular protein delivery: (**A**) schematic representation of CD derivatization with cationic ligands; (**B**) formation of protein-CD nanoassembly; (**C**) intracellular distribution of protein-CD complexes; (**D**) molecular modelling of interactions between nuclear transported importin and CD ligands; (**E**) scheme of intracellular uptake and transport of CD-protein complex. Reproduced with permission from [83]. Copyright 2022 Elsevier.

**Figure 8 ijms-25-09516-f008:**
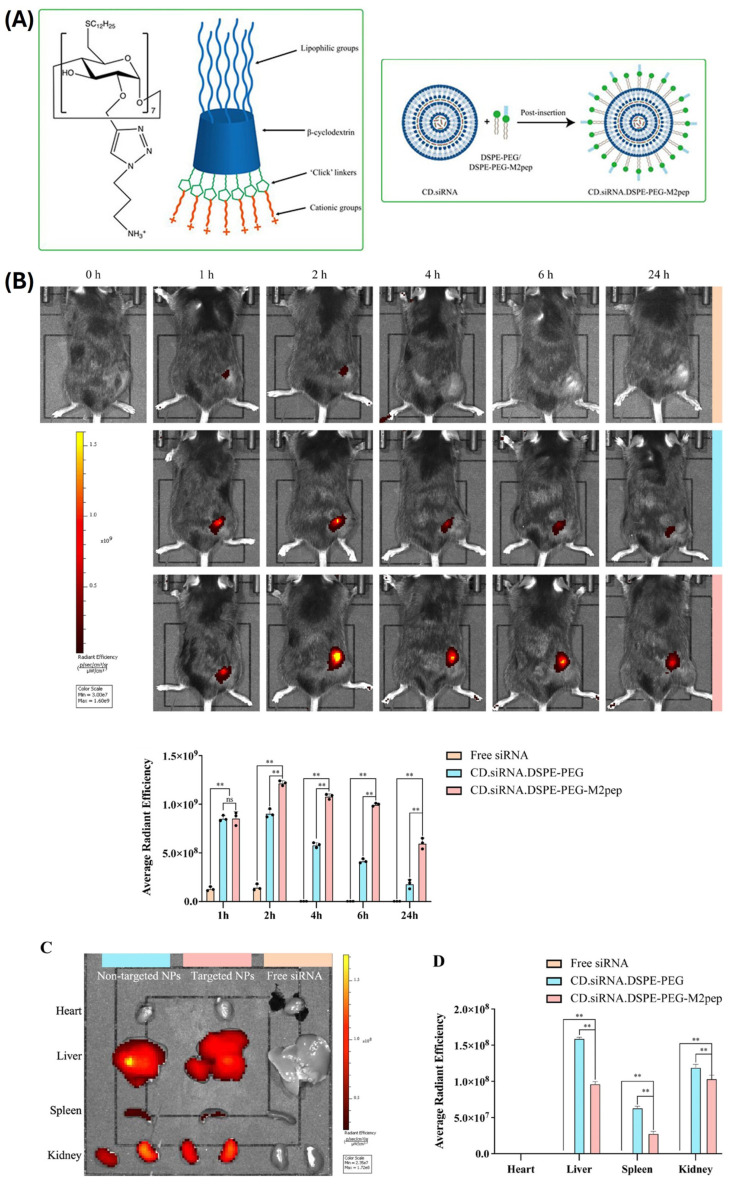
Vesicle-shaped self-assembly of amphiphilic NH_2_Pr-CD-C12 for intracellular siRNA delivery: (**A**) Structure of amphiphilic CD and derivatization of formed vesicles; (**B**–**D**) in vivo biodistribution of siRNA-loaded NH_2_Pr-CD-C12. Reproduced with permission from [93] under the CC-BY 4.0 license. ** *p* < 0.01.

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
