# Peer review of "The Many Faces of Cyclodextrins within Self-Assembling Polymer Nanovehicles: From Inclusion Complexes to Valuable Structural and Functional Elements"

_ijms, 2024, doi:10.3390/ijms25179516_

Round 1

Reviewer 1 Report

Comments and Suggestions for Authors

The review proposed on the use of cyclodextrins as nanovectors describe the state of the art of the research in this field. The review appears accurate and acceptably clear. The analysed  examples are many and cover the most used approaches to the  modification of cyclodextrins ( I would suggest to use the plural cyclodextrins in the title). As a chemists I would have chosen  a deeper look at the chemical elaboration of the structure and use a larger amount of reaction schemes but I beleive that this is a just a matter of style and accept the choice of the authors. The review might be published as it is after a check at the scattered grammar mistakes and missing words.  (for example "polyrotaxene" sholu be "polyrotaxane on page 2 line 46, the sentence "similar to that an aqueous ethanolic solution " on line 85 do notvreally give any info; line 95 misses a word; lines 96 and 97 sounds odd: is NMR useful to detrmine the solubility of a compound?; line 110  repetition of a concept; line 113 an unimolecular micelle simply cannot disassemble upon dilution, it is  not a problem of being resistant....and so on)

Comments on the Quality of English Language

The text should be revised to amend the scattered mistakes and missing words. The clearness of the presentation is acceptable

Author Response

Reviewer 1:

The text should be revised to amend the scattered mistakes and missing words. The clearness of the presentation is acceptable

Answer:

We thank the Reviewer for his observations and recommendations. We have carefully read the manuscript and have revised the text as advised.

Additionally, we have rephrased the phrase about the role of the NMR in solubility elucidation (labelled in cyan). It now reads:

NMR and diffraction experiments demonstrated that the high crystal lattice energy and intramolecular hydrogen bonds prevent CDs from interacting with the neighboring water molecules and render all natural CDs only partially soluble in water

Reviewer 2 Report

Comments and Suggestions for Authors

This is a very nice review, which covered all the aspects of cyclodextrin formulation comprehensively. The figures effectively illustrated the key points to be conveyed! I do not have further comments for changes.

Author Response

Dear Editor,

Thank you for your reply.

Reviewer 3 Report

Comments and Suggestions for Authors

In the review titled: “Many faces of cyclodextrin within the self-assembling 2 nanovehicles: from inclusion complexes to valuable structural 3 and functional element” is reviewing the most recent advances in the design of cyclodextrins self-assemblies and their use for delivery of a wide range of therapeutic molecules.

In the present review, is discussed the possibilities of using new types of controlled release systems based on nanosized self-assemblies of cyclodextrins enabling to control release of different therapeutic molecules. It reveals the many roles of cyclodextrins as drug nanocarriers as well as current challenges and future perspectives.

Introduction is well written and focused on the potential of CD for delivery of diverse classes of therapeutic molecules. Sequentially are described general considerations on amphiphilic cyclodextrins including structure, chemical properties, subtypes and derivatives of CDs with improved solubility.  After that is described the different drug delivery systems, following by section for peptide and protein delivery, nucleic acid delivery

Results are well illustrated with high quality figures, graphs and very informative tables

The final remarks highlighted the broad spectrum of useful properties of CDs as carrier systems and their optimization limits which can reveal the true potential of this class of CD derivatives.

My opinion is that the review can be published in the present form.

Author Response

Dear Editor,

Thank you for your reply.

Reviewer 4 Report

Comments and Suggestions for Authors

The review submitted by Jaraj et al. discuss about the self-assembly of cyclodextrin complexes studied in the last decade. The manuscript is clear, well written and the perspectives proposed by the authors can be taken into account for further studies. However, some corrections are necessary:

1. I think that the authors must also discuss about the complexes between cyclodextrins and amphiphilic copolymers like Pluronics. 

2. line 110: delete "smaller"

3. line 151: replace "polarization" with "polymerization"

Author Response

Reviewer 3:

  1. 1. I think that the authors must also discuss about the complexes between cyclodextrins and amphiphilic copolymers like Pluronics. 

Answer:

Thank you for the suggestion. We have also introduced new paragraphs covering the self-assembling poly and poly(pseudo)rotaxanes. Added text, figure (Fig ) and references are labelled in yellow.

  1. 2. line 110: delete "smaller"

Answer:

Thank you for the observation. It was corrected as suggested.

  1. 3. line 151: replace "polarization" with "polymerization"

Answer:

Thank you for the observation. It was corrected as suggested.